# Prognostic Factors in Therapy Regimes of Breast Cancer Patients with Brain Metastases: A Retrospective Monocentric Analysis

**DOI:** 10.3390/cancers17020261

**Published:** 2025-01-15

**Authors:** Carolin Julia Curtaz, Judith Harms, Constanze Schmitt, Stephanie Tina Sauer, Sara Aniki Christner, Almuth Keßler, Achim Wöckel, Patrick Meybohm, Malgorzata Burek, Julia Feldheim, Jonas Feldheim

**Affiliations:** 1Department of Gynecology and Obstetrics, University Hospital Würzburg, 97080 Würzburg, Germany; 2Department of Diagnostic and Interventional Radiology, University Hospital Würzburg, 97080 Würzburg, Germany; sauer_s3@ukw.de (S.T.S.); christner_s@ukw.de (S.A.C.); 3Section Experimental Neurosurgery, Department of Neurosurgery, University Hospital Würzburg, 97080 Würzburg, Germany; 4Department of Anaesthesiology, Intensive Care, Emergency and Pain Medicine, University Hospital Würzburg, 97080 Würzburg, Germany; 5Department of Neurology, University Hospital Nürnberg, Paracelsus Medical University, 90471 Nürnberg, Germany

**Keywords:** brain metastases in breast cancer, prognostic factors, metastatic breast cancer

## Abstract

Breast cancer patients who develop brain metastases (BMs) face high mortality rates and significant declines in quality of life. Despite its prevalence, knowledge about prognostic factors for brain metastases in breast cancer (BMBC) remains limited. This study analyzed data from 337 BMBC patients treated at our institute between 2004 and 2021. Findings revealed that regional lymph node involvement at the initial diagnosis, the triple-negative breast cancer subtype at BM onset, and extracranial metastases (bone, liver, lung) at the time of BM diagnosis were linked to worse prognoses. Conversely, the HER2/neu-positive subtype, a single BM, BM resection, and stereotactic radiotherapy were associated with prolonged survival. The number of therapies before BM diagnosis also influenced prognosis. Along with known predictors like BM count, extracranial metastases, and treatment modalities, factors such as prior therapies and initial lymph node status could be used in future prognostic models.

## 1. Introduction

On average, one in eight women will experience a breast cancer disease during life. Despite significantly improved breast cancer mortality figures, approximately 10–15% of all patients with breast cancer (BC) and 5–40% of all patients with metastatic BC, including leptomeningeal CNS metastases, develop brain metastasis (BM), one of the most severe types of metastases which is associated with an abysmal prognosis of only 20% one-year survival [1,2,3,4,5].

Breast cancer is the second most common cause of CNS metastases after lung cancer [6]. Parenchymal CNS metastases at autopsy are found in 30–40%, while leptomeningeal CNS metastases in 5–16% of patients. The growing incidence is related to more effective treatment of extracerebral sites with consecutively prolonged survival and increased reporting due to diagnostic methods like cMRI (cranial magnetic resonance imaging) [7,8,9].

Still, there is a lack of specific knowledge about the treatment of brain metastases in breast cancer since most studies of brain metastases are not breast cancer-specific, and pivotal drug trials exclude patients with BMBC. Only recently has this patient population become a research focus [10]. Moreover, for many substances used in metastatic breast cancer (MB), it is entirely unclear whether they can cross the blood–brain barrier (BBB) or not [10]. In short, drug permeability at the BBB is unclear for almost all common substances used to treat metastatic breast cancer. Basic research in permeability changes at the BBB after treatment with drugs or the intracerebral uptake of these drugs is insufficient. Information on the effectiveness of anti-cancer agents in treating brain metastases (BMs) is scarce for various reasons [11,12]. Besides the fact that changes in the expression of molecular receptors have been frequently observed in BMs compared with primary BC [13,14], human tissue samples of brain metastases (BMs) for translational research are limited in availability [12]. It is unclear what effects long-established or newly approved drugs have on the BBB. Do they overcome the BBB and contribute to the treatment of cerebral metastases, or do they have a detrimental effect and reduce the protective function of the BBB by increasing its permeability? It was initially assumed, for example, that trastuzumab therapy in HER2-positive patients leads to increased cerebral metastasis [15], until it finally emerged that the molecular factor of HER2-positive patients leads to increased formation of cerebral metastases [9,10,11]. To date, the effect of new drugs on the BBB has played no or only a minor role in their approval. In almost all studies, patients with cerebral metastasis are excluded [16]. The data situation is, therefore, very limited [17,18]. In addition to the size of the respective molecules, their composition (lipophilic vs. hydrophilic), cerebral transport systems, and the interaction of these factors also play a role [19]. With regard to the effect of drugs on the BBB and cerebral metastasis, there are hardly any evaluable studies in breast cancer. This issue has only recently become the focus of scientific attention [20]. The effects of the chemotherapeutic agents used on the BBB due to their toxicity is a field with large gaps in knowledge.

The highest number of cerebral metastasis is seen in patients with HER2/neu-positive breast cancer and triple-negative breast cancer (TNBC) [7]. It has been shown that there is often a discordance or receptor switch of molecular subtype between primary tumor and BMs [14]. There is no evidence for a survival benefit of BM screening in asymptomatic BC patients [21,22].

For the treatment of single or solitary brain metastasis and oligo-brain metastases, current guidelines and therapeutic standards recommend resection and irradiation of the tumor bed [23,24,25].

The histologic subtype should be considered for the systemic therapy of brain metastases. In recent years, promising progress has been achieved in optimizing therapy regimens, particularly in HER2/neu-positive breast cancer [26]. Despite progress in this area, there are still significant gaps in our knowledge of the optimal individualized treatment of BMBC patients.

Therefore, we aimed to identify (1) characteristics that might be associated with BMBC patients’ survival or even offer prognostic value, as well as (2) therapeutic choices that might be positively/negatively associated with patients’ survival after a primary diagnosis of BMs, depending on the respective subgroups.

## 2. Materials and Methods

In this monocentric retrospective study, we screened all patients treated in the Department of Gynecology and Obstetrics, University of Würzburg, Würzburg, Germany, between 1 January 2004 and 31 December 2021. The data were analyzed in accordance and after consultation with the Würzburg ethics committee in compliance with legal requirements, professional ethics aspects, and applicable data protection regulations.

Patients were included in the trial if they had been diagnosed with BC and one or multiple BMs (at any point during the clinical course). We included patients regardless of the subtype of BC or BM (e.g., only leptomeningeal) and independent of whether the initial diagnosis of either BC or BM had been made at our hospital or whether patients had been referred to our hospital at a later stage of treatment. A diagnosis-related search of internal databases identified the patients screened for eligibility by two independent investigators (CC, JH).

We retrospectively collected basic demographic and clinical data obtained within the framework of routine clinical assessment, focusing on general demographic values, histological and clinical data, patient outcomes, and the number and type of treatment regimens (chemotherapy, radiation, and surgical therapy) before and after the diagnosis of BM. We performed statistical analyses with IBM SPSS Statistics 28 (IBM Corporation, Armonk, NY, USA). Demographic data are either shown in quantity and percentages or median and quartiles. Patients’ survival was compared with Kaplan–Meier analyses (log-rank-test). Median survival and the percentiles (25% and 75%) were calculated in days. However, median survival is also given in months for better understanding. The variables identified as associated with patients’ survival by Kaplan–Meier analyses (HER2/neu, TNBC, extracranial metastasis in general, bone metastasis, liver metastasis, lung metastasis, skin metastasis, lymph node metastasis, singular/multifocal BMs, surgery of BMs, radiotherapy of BMs, lines of therapy before BMs, involvement of regional lymph nodes determined by the N-classification) were included into a multivariable stepwise backward Cox regression for overall survival after BMs (exclusion criteria *p* > 0.05).

## 3. Results

### 3.1. Patients’ Characteristics

Detailed data of the patients’ characteristics and demographic data, as well as tumor histology, can be seen in Table 1. The median age at BC diagnosis was 51 years (quartiles: 44–60 years), while the median age at BM diagnosis was 57 years (quartiles: 49–66 years). The most common tumor size was T2 (41.5%), and the least common was T4 (12.2%). Lymph node metastases were present in 57.3% of patients, with N3 being the least frequent at 12.4% (42 patients). Most patients with BMs initially had a BC diagnosis with a G3 grading (55.4%/150).

A total of 22% of patients were classified with the triple-negative breast cancer (TNBC) subtype, while HER2/neu positivity was observed in 39.4%. Among HER2/neu-positive patients, 53.5% were also hormone receptor-positive. Since Ki67 was not consistently evaluated in patients treated before 2010, differentiating between luminal A and B subtypes was impossible based on the available data.

Ductal carcinoma was the classification for 56.4% of tumors, whereas the lobular subtype accounted for only 8.0%. Multifocal brain metastases (BMs) were detected in 69.4% (234 patients) compared to 19.0% (64 patients) with a single BM. Leptomeningeal metastases (LMs) were identified in 7.4% (25 patients). We classified meningeosis according to the EANO ESMO classification [5], with Type I, classified as confirmed LMs, diagnosed in 7.7% (26) of the cohort; Type II, classified as probable LMs, diagnosed in 9.5% (32); and Type III, classified as possible LMs in 0.3% (1) of the cohort.

About three-fourths of the patients with later-diagnosed BMs received no neoadjuvant therapy (77.7%/262). About half of the patients (49.9%/168) were treated with therapies against bone absorption. The following data were obtained about the initial surgical treatment of breast cancer: about 50.4% (170) received a breast-conserving therapy, and 49% received a mastectomy (165).

The general staging of metastatic breast cancer does not include a standardized cMRI. In our patient cohort, 78% (268) patients experienced neurological symptoms, while 6.2% (21) had no symptoms, and for 15.7% (53), no data were available.

The surgical resection of BMs was not performed in 76.6% of patients, while 15.4% (52 patients) underwent tumor-reducing brain surgery. Whole-brain radiotherapy was administered to 58.2% (196 patients), 8.9% (30 patients) received stereotactic radiotherapy, and 16.6% (56 patients) underwent a combination of whole-brain and stereotactic radiotherapy.

Radiotherapy was not administered to 8% (27) of the patients. More than half of the cohort (58.2%/196) was treated with whole-brain radiotherapy of the BMs, while 8.9% (30) were treated with stereotactic radiotherapy and 16.6% (56) received combined whole-brain and stereotactic radiotherapy.

### 3.2. Survival Analyses Depending on the Receptor State

The overall cohort’s median survival after diagnosis of BM was 7.0 months (212 days; quartiles 69–537 days). Stratified into the histological subtypes, BM patients in the HER2/neu-positive cohort survived a median of 16.6 months (414 days; quartiles 105–860 days) compared to 5.4 months (164 days; quartiles 50–443 days) in the luminal-like subgroup and 4.3 months (132 days; quartiles 56–338 days) in the TNBC subgroup (log-rank test *p* < 0.001; Figure 1).

Overall survival (OS), classified as the time from diagnosis BC to death, was 65.5 months (1991 days, quartiles 1038–4207 days). Progression-free survival (PFS), defined as the time from primary diagnosis of BC to the diagnosis of BMBC, was analyzed as 45.1 months (1371 days; quartiles 685–2869 days) (referring to Table 1).

### 3.3. Negative Prognostic Factors

#### 3.3.1. Involvement of Regional Lymph Nodes at Primary Diagnosis of BC and Subtype of TNBC

In Kaplan–Meier analyses (Figure 2), an increased involvement of regional lymph nodes (N-classification) at the primary diagnosis of BC correlated with poorer overall survival from the time of diagnosis of BM to death (Log-rank test *p* = 0.026). Patients with N0/N1 at primary diagnosis displayed a median survival from BM to death of 7.4 months (224 days; quartiles 75–539 days), N2 of 8.1 months (247 days; quartiles 71–631 days), and N3 of 3.9 months (118 days; quartiles 49–360 days). N3, in particular, is therefore an adverse prognostic factor. We analyzed N0 and N1 as a combined group, since the N0 subgroups only consisted of five individuals, and clinically, the operative standard of surgical axillary dissection in BC patients is becoming less invasive [16].

#### 3.3.2. Extracranial Metastasis at the Time of Brain Metastasis Diagnosis

Many patients (82.5%/287) displayed extracranial metastases at the time of initial diagnosis of BM. Extracranial metastases at the time of BM diagnosis were negatively associated with the survival time (median survival: 5.6 months [170 days; quartiles 63–507 days] vs. 10.9 months [332 days; quartiles 113–900 days] without extracranial metastasis; log-rank test *p* = 0.013).

Thus, the presence of extracranial metastases at the time of BM diagnosis can be listed among the adverse prognostic factors. Next, we analyzed how the different types of extracranial metastases at the time of BM diagnosis affect patients’ survival.

Bone metastases were observed in 59.3% (200 patients), while lung metastases were present in 45.7% (154 patients) (Figure 3). Liver metastases affected 40.9% (138 patients), lymph node metastases occurred in 33.5% (113 patients), and skin metastases were reported in 8.3% (28 patients). The presence of bone metastases at the time of BM diagnosis showed a statistically non-significant tendency towards poorer survival (5.2 months [158 days; quartiles 54–513 days] vs. 9.0 months [273 days; 104–558 days] without bone metastasis; log-rank test *p* = 0.075) (Figure 4A).

In contrast, liver metastases at the time of BM diagnosis were strongly associated with overall survival in our patient collective (median 4.0 months [122 days; quartiles 47–371 days] vs. median 10.4 months (315 days; quartiles 108–631 days; log-rank test *p* = 0.001)) (Figure 4B).

Lung metastases at the time of BM diagnosis also showed a significant negative impact on the survival of BMBC patients (4.9 months/149 days; percentiles 65–486 days vs. 8.9 months/272 days; percentiles 88–672 days without lung metastases; log-rank test *p* = 0.001) (Figure 4C).

### 3.4. Positive Prognostic Factors

#### 3.4.1. Immunohistochemical Subtype of HER2/Neu-Positive BMBC

Of the 337 patients, 275 were analyzed for HER2/neu status and overall survival. We observed that the overexpression of HER2/neu in Kaplan–Meier analysis led to a clear survival advantage from the diagnosis of BM until death (13.6 months [414 days; quartiles 105–860 days] vs. 4.9 months [149 days; quartiles 56–370 days]; log-rank test *p* < 0.001; Figure 1).

#### 3.4.2. Singular Brain Metastasis

We obtained data on the localization and number of BMs (singular vs. multifocal vs. only leptomeningeal) for 263 patients. The occurrence of a singular BM was associated with the most favorable overall survival (singular BM: median survival 13.5 months [411 days; quartiles 100–997 days] vs. multifocal BM median survival: 6.4 months [194 days; quartiles 69–463 days] vs. only leptomeningeal growth median survival 4.7 months [142 days; quartiles 46–596 days]; log-rank test *p* < 0.001).

#### 3.4.3. Surgery of Brain Metastasis

Patients who received metastasis resection (15.4%/52) lived significantly longer (median 16.0 months [488 days; quartiles 231–1536 days] vs. 5.4 months [164 days; quartiles 69–513 days]) than patients who did not undergo surgery (log-rank test *p* = 0.001). This effect was also prevalent in the subgroup of patients with singular metastasis (*p* = 0.002).

#### 3.4.4. Radiotherapy of Brain Metastasis

In addition to surgical therapy, we analyzed the different modalities of radiotherapy used in BMs (Figure 5). Three hundred and nine patients could be included in these subgroup analyses. The patients were either not subjected to radiotherapy (8.7%/27), treated by whole-brain radiotherapy (63.4%/196), stereotactic radiotherapy (9.7%/30), or whole-brain and stereotactic radiotherapy (18.1%/56) of BMs. Patients who received stereotactic radiotherapy of BMs alone lived the longest, with a median of 20.6 months (627 days; quartiles 144–987 days) (Figure 6). Patients who received both whole-brain and stereotactic radiotherapy of BMs presented a median overall survival of 16.9 months (513 days; quartiles 226–860 days). Patients who received whole-brain radiotherapy of BMs alone survived a median of only 5.4 months (163 days; quartiles 73–443 months), whereas patients without radiotherapy at all only survived 3.2 months (97 days; 19–295 days) after diagnosis of BMs (log-rank test *p* < 0.001).

#### 3.4.5. Lines of Therapy Regimens Before the Diagnosis of Brain Metastasis

Due to the improved response to treatment regimens, more patients receive more lines of therapy (LOTs). Therefore, we analyzed the number of treatment regimens applied before and after the diagnosis of BM (Figure 7). In the analysis of treatment regimens before the occurrence of BM, 14.7% (40) did not receive any chemotherapy, as BM was already diagnosed at the time of BC diagnosis. A total of 33.6% (91) of all patients received one line of chemotherapy before BM was diagnosed. Additionally, 28% (76) of the patient population received two lines of chemotherapy before BM was diagnosed. A total of 12.1% (33) of patients with available data received two lines of chemotherapy before BM was detected. Lastly, 11.4% (31) of the patients received more than three LOTs before BM occurred.

After the diagnosis of BM, 52.2% (142) received no further chemotherapy. After the diagnosis of BM, 27% (74) received only one other chemotherapy regime. A total of 13.2% (36) of patients received two further lines of chemotherapy, and 5.1% (14) of the patients even received three treatment regimens after diagnosis of BM.

The number of therapy regimens before the diagnosis of BM also appears to be associated with patients’ survival after BM. Up to three treatment regimens before the diagnosis of BM seem to be prognostically favorable (no therapy before BM diagnosis median survival: 9.7 months [295 days; quartiles 75–809 days]; one therapy regimen before BM median survival: 8.3 months [252 days; 89–590 days]; two therapy regimens median survival: 7.4 months (226 days; quartiles 96–526 days); three therapy regimens median survival: 4.8 months (145 days; quartiles 56–370 days); more than three therapy regimens median survival: 1.9 months (58 days; quartiles 29–294 days) after BM diagnosis (log-rank test *p* = 0.001) (Figure 8). Therefore, no more than three treatment regimens before the diagnosis of BM appear to be prognostically advantageous.

#### 3.4.6. Multivariable Survival Analysis

To determine possible mutual interferences between the characteristics, we included the factors we identified as positively or negatively associated with survival after primary diagnosis of BM into a multivariable stepwise backward Cox regression. Despite a HER2/neu and triple-negative receptor state, surgery and radiotherapy of BM appeared to be independent predictors for patients’ survival. Also, a regional lymph node involvement of N0/N1 and the existence of extracerebral metastases in general, as well as specifically liver metastases, showed statistical significance in the final model (all *p*-values < 0.05). Different types of extracranial metastases (as well as the other variables considered) were excluded from the model when the general existence of extracerebral metastases was considered.

## 4. Discussion

It is known that breast cancer is the second most common cause of CNS metastases and shows a high incidence in data analyses based on the autopsy of metastatic BC patients (parenchymal CNS metastases: circa 30–40%; leptomeningeal CNS metastases: circa 5–16%) [7,27,28].

The increasing incidence (10% to 40%) of BM in BC patients may be caused by the more effective treatment of extracerebral localizations with an overall improved prognosis, as well as by the increased reporting due to the use of MRI for diagnostic evaluation. Nevertheless, there is still a lack of specific knowledge about the treatment of brain metastases in breast cancer, as most studies are not breast cancer-specific [18]. In addition to a tumor cell’s genetic predisposition, its capacity to cross the blood–brain barrier and adapt to the new microenvironment may also influence its ability to metastasize to the brain [18]. Both metastatic tumor cells and chemotherapy targeting brain metastases must bypass the blood–brain barrier (BBB) to reach the brain. Conversely, breast cancer chemotherapies that aim to target brain metastases can also alter the properties of the BBB, though these effects remain largely unexplored [29]. Patients with active and progressing brain metastases have historically been excluded from clinical trials. This exclusion is primarily due to concerns about increased toxicity, frequent corticosteroid use, prior radiation therapy (RT), impaired performance status, and limited life expectancy [16].

Retrospective data analysis and health services research can help to improve relevant issues around the care of this patient population in the future. Another issue commonly discussed is that it is still challenging to predict the prognosis of BC patients with BM.

Similarly to previous observations, our study population’s average age at the first diagnosis of BMBC was 57 years [8,30,31]. Subgroup analyses of the immunohistological subtypes also showed very similar results. Only the proportion of HER2/neu-positive patients was higher in our cohort than previously reported [30]. The median overall survival (OS) after the development of BMs for the overall cohort was 7.0 months, similar to the data from others [8].

Our retrospective data analysis identified adverse prognostic factors for survival time at BM diagnosis; the immunohistochemical subtype of TNBC and extracranial metastasis at the time of BM diagnosis were negative prognostic markers for overall survival after BM diagnosis. Our data also showed results (4.3 months) similar to previous observations regarding the survival time for patients with TNBC [8,32]. This clearly shows the importance of consistent and successful treatment of localized early TNBC to prevent metastasis in the course of the disease.

Extracranial metastases at the time of BM diagnosis play an essential role in the course of the disease. We could show that existing metastasis to the lymph nodes, lungs, and liver significantly reduced overall survival after BM diagnosis, while bone metastasis had no significant effect on survival. However, it should be noted that the statistical evaluation analyzed the individual metastasis sites; multiple metastasis sites were not taken into account in this observation. This will be a prospective component of further assessments. However, we identified a factor already known at BC diagnosis that plays a role in overall survival after BM: the lymph node status at the initial BC diagnosis. As seen in Figure 2, the risk of poor survival after diagnosis of BM significantly increases with the stage of lymph nodal metastasis at primary diagnosis of BC. In our analyses, lymph nodal metastasis rated as N3 at the initial diagnosis (ten or more lymph nodes in the armpit or under/above the collarbone affected) shows a negative prognostic effect after BM diagnosis.

Different studies have inconsistently assessed the presence of N ≥ 2 (defined as four to nine lymph nodes affected in the axilla) at BC diagnosis. Tham et al. argue that they play a relatively minor role [23]. Shen et al. showed that around 2/3 of all patients with brain metastases had lymph node metastases at initial diagnosis. However, their work did not categorize the number of metastases [33].

This underlines once again that the early detection of localized breast cancer, preferably without lymph nodal metastasis, is the best primary prevention to avoid BMBC; therefore, breast cancer screening yields vital importance.

The longest survival rates were seen in the cohort of BMBC with the immunohistochemical subtype of HER2/neu-positive (Figure 1). Other studies have also demonstrated this effect [8,30,34]. However, the exact reasons have not been finally understood. One potential explanation for this could be that the systemic HER2/neu-based antibodies used in the treatment of BMBC partially cross the BBB and can thus directly influence tumor growth intracerebrally [10]. This known finding led just recently to the publication of an expert consensus on the prevention of brain metastases in patients with HER2-positive breast cancer [35]. However, the steering committee also highlights a lack of high-quality clinical trials that include patients with brain metastases and evaluate various treatments and outcomes. They suggested that such studies, combined with existing consensus statements, could support the development of more comprehensive and updated clinical guidelines in the future [35].

For our study too, most of the available data date back to when the new innovative drugs available today had not yet been approved, and patients were not yet able to receive them [36,37]. This is currently changing and will continue to be the focus of further approval studies of newly established drugs, such as the recently published data on trastuzumab deruxtecan in HER2-positive advanced breast cancer with or without brain metastases [38] or also the treatment with tucatinib as part of the HER2CLIMB study [39]. In addition, there is still insufficient knowledge about the effects on drug permeability at the BBB and intracerebral uptake, as discussed above [10]. In principle, the primary recommendation is to resect cerebral metastases and apply stereotactic radiotherapy if possible. However, this recommendation is complemented by the constant progress in the approval of systemic therapeutic agents and must therefore always be re-evaluated with each new substance in the future. Nationally in Germany, for example, the AGO Mamma deals annually with new recommendations on treatment strategies for BMBC [40], while internationally, there are always new guidelines and directives, such as the aforementioned expert consensus and guidelines, e.g., from ASCO-SNO-ASTRO and EANO-ESMO [25,35,41].

As positive prognostic factors, we identified the occurrence of a singular BM and the surgical removal of BM (Figure 3). Our findings underline the existing recommendations. Local therapy (surgery, stereotactic radiosurgery) depends on localization, size, number of metastases, previous treatment, Karnofsky performance scale, and the general prognosis and should be carried out if possible [37,42]. However, it is important to notice that our retrospective analyses do not consider the patient profile associated with the therapeutic decisions, which largely influences the outcome (e.g., whole-brain radiotherapy is mainly applied in patients with multiple metastases that have a poor prognosis). Our data show that stereotactic irradiation alone achieves better overall survival in BMBC patients compared to whole-brain radiation. This might be due to stereotactic irradiation alone, usually being only used for a single BM or few metastases. Still, both show significant benefits over no radiotherapy at all (Figure 3).

The lines of therapies (0–4) before the diagnosis of BM were also identified as prognostically favorable (Figure 7). One possible explanation could be that more specific and more effective therapeutic agents have been approved in recent years, and the patients analyzed here with a date of death up to 2021 could not benefit from them [43].

Recently, so-called breast GPA (Graded Prognostic Assessment) scores to estimate survival from brain metastases (BMs) have been taken more and more into account [5,30]. In 2008, Sperduto et al. published the Graded Prognostic Assessment (GPA) score for the first time [44]. In 2012, the score was expanded to include the immunohistological subtypes of the primary tumor but also consider prognostic factors such as age and Karnofsky performance [45]. In 2020, this version was updated again, and the factors of extracranial metastases (ECMs) and the number of BMs were added [46]. Our data analysis also verifies most of the factors listed in the scores as prognostic markers. However, the number of initial treatments before BM has not yet been taken into account in the assessments and could be another asset to improve the score, thus better assessing the prognosis of patients and advising them appropriately.

In summary, in addition to established factors such as immunohistologic subtype, number of brain metastases (BMs), surgical resection, stereotactic radiation, and type of extracranial metastases, we also identified the number of therapies prior to BM diagnosis and initial lymph node status as novel prognostic indicators. Apart from the fact that the treatment of brain metastases requires a multidisciplinary strategy that includes multimodal treatments tailored to the clinical and radiological context, our results could influence current clinical practice if they are prospectively evaluated and added, for example, to the breast GPA (Graded Prognostic Assessment) score for estimating survival of BM, which could help affected patients and treating specialists to find individualized treatment regimens.

## 5. Conclusions

Retrospective data analysis is crucial for identifying risk factors influencing survival after a BM diagnosis. In our study, in addition to known factors such as immunohistologic subtype, number of BMs, surgical removal and stereotactic irradiation, and type of extracranial metastasis, we were able to identify the number of therapies before the diagnosis of BM and the initial lymph node status as additional new prognostic factors. These results could potentially influence future clinical practice by incorporating these newly identified factors into, for example, the breast GPA (Graded Prognostic Assessment) scores to assess the survival rate of brain metastases (BMs) and provide appropriate treatment recommendations for affected patients.

## Figures and Tables

**Figure 1 cancers-17-00261-f001:**
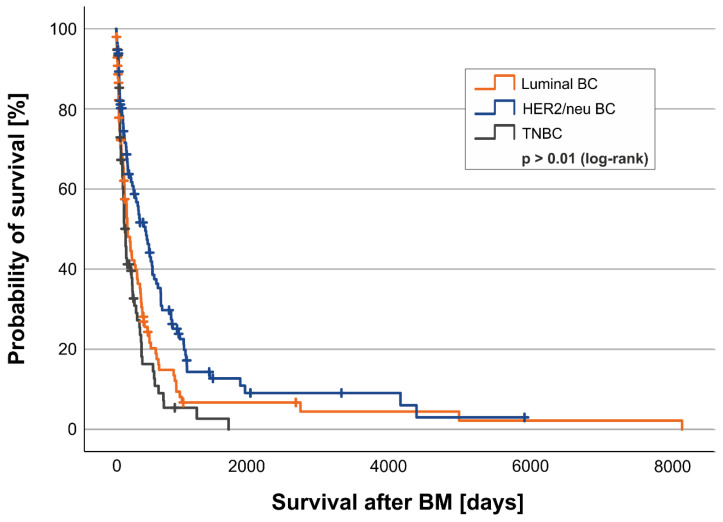
Kaplan–Meier curve for the time from first diagnosis of BM to death due to any reason analyzed by histological subgroups. Shown is the probability of survival [y-axis] over time in days [x-axis] of patients with triple-negative breast cancer (TNBC, gray), HER2/neu-positive breast cancer (HER2/neu BC, blue), and luminal breast cancer (luminal BC, orange). Subgroups were tested for significant differences in survival with the log-rank test.

**Figure 2 cancers-17-00261-f002:**
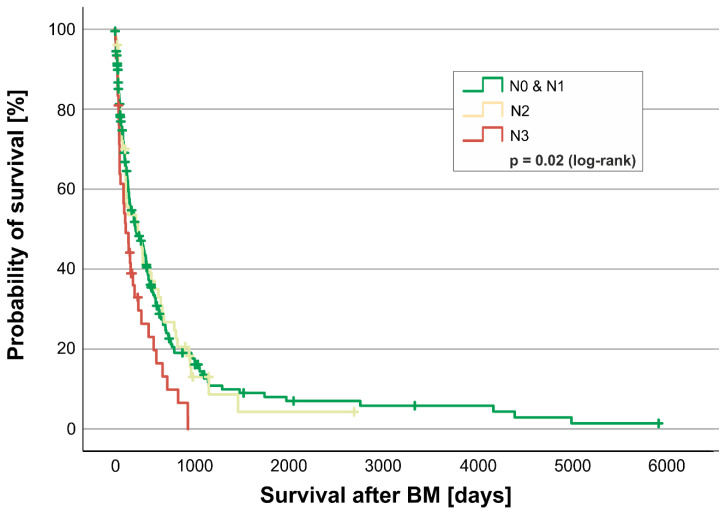
Kaplan–Meier curve for the involvement of regional lymph nodes (N-classification) from primary diagnosis of BM to death for any reason. Shown is the probability of survival [y-axis] over time in days [x-axis] of patients with a lymph node involvement at diagnosis classified as N0/N1 (green), N2 (yellow), and N3 (red). Subgroups were tested for significant differences in survival with the log-rank test.

**Figure 3 cancers-17-00261-f003:**
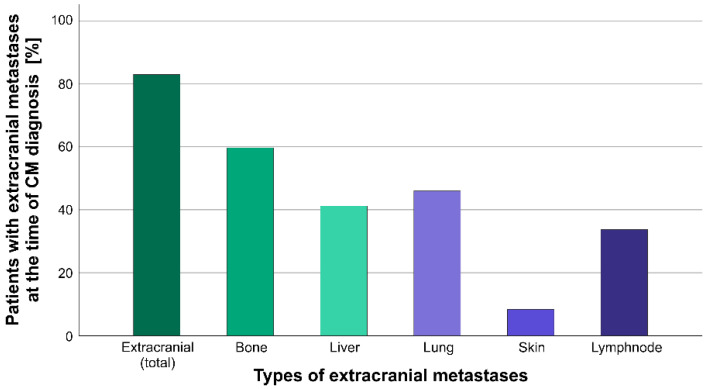
Bar graph displaying the occurrence of extracranial metastases (in total, bone metastases, liver metastases, lung metastases, skin metastases, and lymph node metastases) at the time of cerebral metastasis (CM) diagnosis in percentages [%].

**Figure 4 cancers-17-00261-f004:**
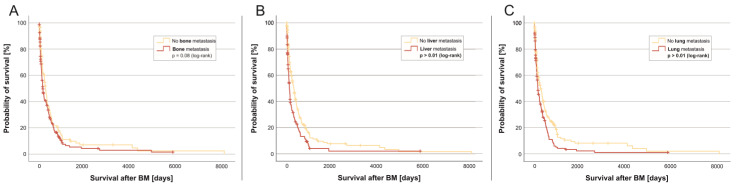
Kaplan–Meier curve for different metastatic sites at the time of brain metastasis diagnosis. (**A**) Kaplan–Meier curve for the involvement of bone metastases from primary diagnosis of BM to death due to any reason; (**B**) Kaplan–Meier curve for the involvement of liver metastases from primary diagnosis of BM to death due to any reason; (**C**) Kaplan–Meier curve for the involvement of lung metastases from primary diagnosis of BM to death due to any reason. The probability of survival [y-axis] over time in days [x-axis] is shown. Subgroups were divided by whether patients were (red) or were not (yellow) diagnosed with the specific type of metastasis at the time of brain metastasis diagnosis and were tested for significant differences in survival with the log-rank test.

**Figure 5 cancers-17-00261-f005:**
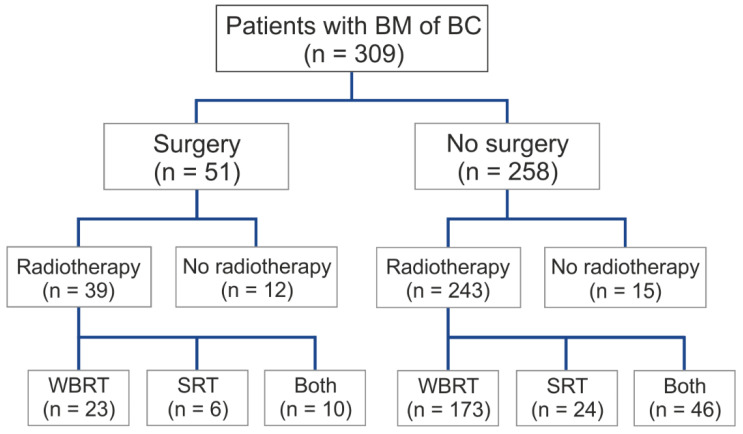
Local treatment after the diagnosis of BM; BM, brain metastasis; WBRT, whole-brain radiotherapy; SRT, stereotactic radiotherapy.

**Figure 6 cancers-17-00261-f006:**
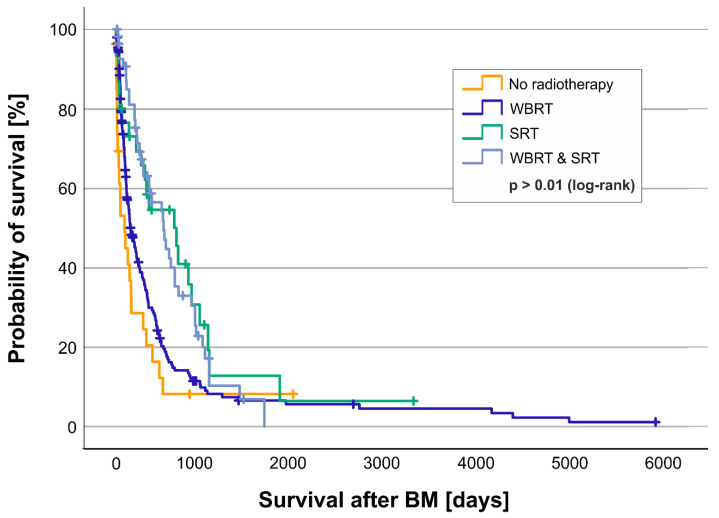
Kaplan–Meier curve for the involvement of different types of radiotherapy of BMs. Shown is the probability of survival [y-axis] over time in days [x-axis] of patients not treated with radiotherapy (yellow), treated with whole-brain radiotherapy (dark blue, WBRT), stereotactic radiotherapy (green, SRT), or combined whole-brain and stereotactic radiotherapy (light blue, WBRT and SRT). Subgroups were tested for significant differences in survival with the log-rank test.

**Figure 7 cancers-17-00261-f007:**
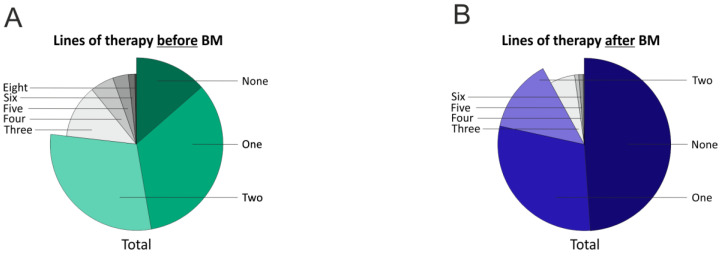
Pie chart showing the number of lines of therapy in patients with BM; before diagnosis of BM (**A**) and after diagnosis of BM (**B**).

**Figure 8 cancers-17-00261-f008:**
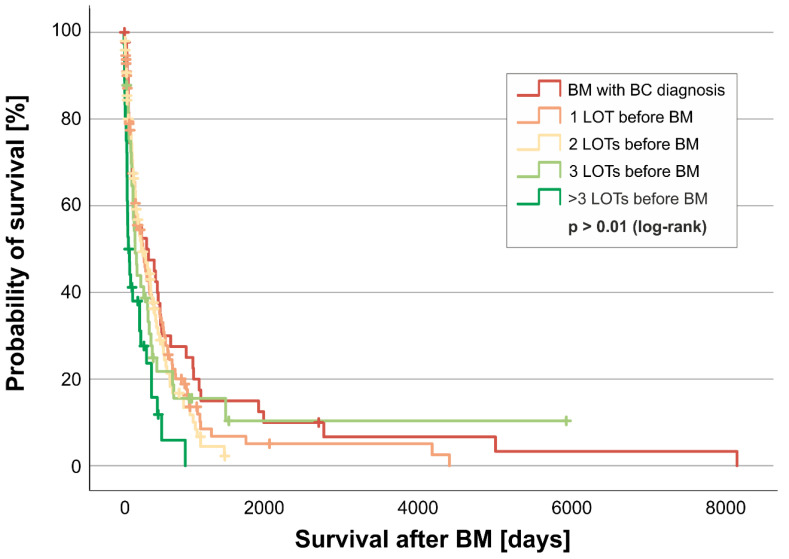
Kaplan–Meier curve for the line of therapy before the diagnosis of BM; LOT = line of therapy. Shown is the probability of survival [y-axis] over time in days [x-axis] of patients that received zero (dark red), one (light red), two (yellow), three (light green), or more than three (dark green) lines of therapy before breast metastasis diagnosis. Subgroups were tested for significant differences in survival with the log-rank test.

**Table 1 cancers-17-00261-t001:** Clinical parameters of breast cancer patients. The absolute numbers of breast cancer patients in each group and the percentage of the analyzed population or the median and quartiles are given.

Demographic Data and Tumor Histology				
**Age at diagnosis of BC (median)**	51 years (44–60 years)			
**Age at diagnosis of CM (median)**	57 years (49–66 years)			
**Tumor size** **(T-classification)**	T0: 1/0.3%	T1: 102/30.3%	T2: 140/41.5%	
	T3: 30/8.9%	T4: 41/12.2%	Unknown: 23/6.8%	
**Involvement of regional lymph nodes (N-classification)**	N0: 100/29.7%	N1: 100/29.7%	N2: 51/15.2%	
	N3: 42/12.4%	Unknown: 44/13.1%		
**Tumor grading** **(G-classification)**	G1: 4/1.2%	G2: 117/34.7%	G3: 150/44.5%	Unknown: 66/19.6%
**Histological subtype**	Luminal: 99/34.2%	HER2-positive: 114/33.8%	Triple-negative: 76/22.6%	Unknown: 48/14.2%
**Tumor histology**	Ductal *: 190/56.4%	Lobular *: 27/8.0%	Others:5/1.5%	Unknown: 115/34.1%
**Therapy**				
**Neoadjuvant therapy**	Neoadjuvant therapy: 68/20.2%		No neoadjuvant therapy: 262/77.7%	Unknown: 7/2.1%
**Therapy against bone resorption**	Therapy against bone resorption: 168/49.9%		No therapy against bone resorption: 167/49.6%	Missing: 2/0.6%
**Surgical intervention after diagnosis of BC**	Only biopsy: 17/5.0%		Breast-conserving resection: 142/42.1%	Mastectomy: 137/40.7%
	Breast-conserving resection followed by mastectomy: 28/8.3%		Unknown: 13/3.9%	
**Recurrence and metastases**				
**Extracerebral metastases of BC before or with the diagnosis of BMs**	Extracerebral metastasis: 278/82.5%		No extracerebral metastasis: 57/16.9%	Unknown: 2/0.6%
**Localization of extracerebral metastases**	Bone metastases: 200/59.3%		Liver metastases: 138/40.9%	Lung metastases: 154/45.7%
	Skin metastases: 28/8.3%		Lymph node metastases: 113/33.5%	
**Karnofsky index at diagnosis of BMs**	10–30%: 5/1.5%		40–50%: 24/7.1%	60–70%: 48/14.2%
	80–90%: 125/37.1%		100%: 60/17.8%	Unknown: 75/22.3%
**Symptoms of BMs**	Neurological symptoms: 263/78.0%		Asymptomatic: 21/6.2%	Unknown: 53/15.7%
**Localization of BMs**	Singular solid metastasis: 64/19.0%		Multifocal solid metastases: 234/69.4%	
	Only leptomeningeal: 25/7.4%		Unknown: 14/4.2%	
**Therapy of BMs—surgery**	Resection: 52/15.4%		No resection: 258/76.6%	Unknown: 27/8.0%
**Therapy of BMs—** **radiotherapy**	Whole-brain radiation **: 252/74.8%		Stereotactic radiation **: 86/25.5%	
	No radiotherapy: 27/8.0%		Unknown: 28/8.3%	
**Survival**				
**OS** **(diagnosis BC—death)**	65.5 months (1991 days, quartiles 1038–4207 days)			
**OS** **(diagnosis BMs—death)**	7.0 months (212 days; quartiles 65–537 days)			
	Luminal-like subgroup 5.4 months (164 days; quartiles 50–443)		HER2-positive 16.6 months (414 days; quartiles 105–860 days)	TNBC 4.3 months (132 days; quartiles 56–338 days)
**PFS (diagnosis BC—diagnosis BMs)**	45.1 months (1371 days; quartiles 685–2869 days)			

* Five patients with histological features of lobular and ductal breast cancer. ** Fifty-six patients received whole-brain and stereotactic radiation. Missing/unknown data are due to several reasons. (1) Patients treated at the beginning of our selection period were partially graded and diagnosed due to older standards, and reclassification was not possible for every patient. (2) Few patients were either diagnosed elsewhere and later referred to our institute, partly treated elsewhere during their clinical course, or lost to follow-up, leading to incomplete information regarding applied therapies (e.g., modality of radiotherapy performed at an external institute) or precise location of metastases/recurrences). (3) Due to incomplete documentation, the Karnofsky index and clinical symptoms at BM diagnosis could not be reconstructed for every patient. Abbreviations: BC, breast cancer; BM, brain metastasis; OS, overall survival; PFS, progression-free survival.

## Data Availability

The datasets analyzed during the current study are available from the corresponding author upon reasonable request.

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
