# Peer review of "Prognostic Factors in Therapy Regimes of Breast Cancer Patients with Brain Metastases: A Retrospective Monocentric Analysis"

_cancers, 2025, doi:10.3390/cancers17020261_

Round 1

Reviewer 1 Report

Comments and Suggestions for Authors

This is an interesting retrospective study addressing the prognosis of of breast cancer patients with brain metástases which can provide relevant information for clinical practice, Congratulations to the authors!

I have a few comments about some aspects of the study:

1) The percentage of BM in BC described in the abstract (15-30%) differs from that presented in the introduction (10-15%) and in the discussion on page 10, line 289 (10% to 40%). It would be beneficial to harmonize these values for consistency.

2) On Page 2, line 83: Should we really refer to the study as a "trial"?

3) On Page 3, lines 114 and 140: It is not recommended to start a paragraph with a number.

4) In the Results section: It is not necessary to present the results for all variables, as they are already shown in Table 1. Therefore, I suggest highlighting only the most relevant values (e.g., the highest or lowest ones).

5) It is evident that patients with HER2+ status have superior survival. Did all of them receive anti-HER2 therapy? This would be an important piece of information to consider, especially regarding access to new therapies.

6) On page 7, line 117, Singular BM is presented as a prognostic factor. Was there a difference in outcomes for patients who received surgery or radiotherapy for a single metastasis?

7) It would be relevant to include in the discussion that it is not the modality of radiotherapy for BM that correlates with a worse prognosis, but rather the patient profile associated with each modality (e.g., whole-brain radiotherapy for multiple metastases).

Author Response

1) The percentage of BM in BC described in the abstract (15-30%) differs from that presented in the introduction (10-15%) and in the discussion on page 10, line 289 (10% to 40%). It would be beneficial to harmonize these values for consistency.

Thank you for pointing this out. We agree with you on the comment and therefore harmonized the values for consistency.

2) On Page 2, line 83: Should we really refer to the study as a "trial"?

Thank you for pointing this out. We agree with you on the comment and rephrased this as retrospective study.

3) On Page 3, lines 114 and 140: It is not recommended to start a paragraph with a number.

Thank you for pointing this out. We agree with you on the comment and rephrased the sentence and also following sentences in the paper not starting with a number.

4) In the Results section: It is not necessary to present the results for all variables, as they are already shown in Table 1. Therefore, I suggest highlighting only the most relevant values (e.g., the highest or lowest ones).

Thank you for pointing this out. We agree with you on the comment and shortened the text by highlighting the most relevant values.

5) It is evident that patients with HER2+ status have superior survival. Did all of them receive anti-HER2 therapy? This would be an important piece of information to consider, especially regarding access to new therapies. (

We thank the reviewer for this important remark. Out of the 114 Her2neu-positive patients, 11 were immediately diagnosed with cerebral metastases, and another 19 did not receive HER2-antibodies before CM, leaving 81 that were treated with HER2 antibodies before CM. After CM, only 44 patients received HER2 antibodies, most commonly because they had been applied before and developed metastases/recurrences under anti-HER2 antibodies. We have not added this information to the manuscript, as we agree with the reviewer that the subgroup of HER2-positive patients deserves special attention and are, therefore, currently working on an extensive subgroup analysis that would, however, exceed the extent of this manuscript.  

6) On page 7, line 117, Singular BM is presented as a prognostic factor. Was there a difference in outcomes for patients who received surgery or radiotherapy for a single metastasis?

We are glad to provide the additional information. Indeed, surgery remained a decisive prognostic factor in the subgroup of patients with a single metastasis. Patients who did not receive surgery survived a median of 252 days after diagnosis of CM, whereas patients who received tumor resection survived a median of 900 days (p>0.01 log-rank). These observations are biased by other factors that led to the selection for surgery. In contrast, radiotherapy was not significantly associated with survival in this subgroup. We have added this information in the subsection “Surgery of Brain Metastasis”.

7) It would be relevant to include in the discussion that it is not the modality of radiotherapy for BM that correlates with a worse prognosis but rather the patient profile associated with each modality (e.g., whole-brain radiotherapy for multiple metastases).

We thank the reviewer for highlighting this omission and have added a remark to our discussion.

Reviewer 2 Report

Comments and Suggestions for Authors

The study entitled "Prognostic Factors in Therapy Regimes of Breast Cancer Patients with Brain Metastases: A Retrospective Monocentric Analysis" provides a comprehensive retrospective assessment of data collected from 337 patients experiencing brain metastases secondary to breast cancer. The employed methodologies are meticulously detailed, demonstrating that variables such as immunohistochemical subtype and initial lymph node status significantly influence survival outcomes. Nonetheless, the study could be enhanced with a more detailed representation of complex data and a thorough discussion regarding the clinical implications of the findings.

Major:

Introduction Section:
1. Lines 50-55, Page 2. It is suggested that the authors expand the background discussion on drug permeability across the blood-brain barrier. For instance, incorporating insights from Corti et al., 2022, may provide a more comprehensive understanding of intracranial efficacy relevant to the treatment of brain metastases.

Materials and Methods Section:
2. Despite well-documented methods, the inclusion of a flowchart could improve comprehension of the participant selection process.

Results Section:
3. Although extensive, the results might be improved by adding additional tables or graphs, especially when presenting complex demographic data. It is essential to clearly justify any missing data in particular categories.
4. Within the Kaplan-Meier plots, it is advisable to clarify the legends, colours, and accompanying statistics to aid result interpretation.
5. Lines 187-194, Page 6. A graphical representation could potentially improve the presentation and interpretation of these results.
6. Page 7, Lines 202-203. Interpreting these results is challenging. Contrary to the textual description, the accompanying figure and caption do not indicate that the values correspond to the brain metastases group.

Discussion Section:
7. The study mentions related works but lacks depth. An expanded account of the referenced studies could enrich the discussion, providing better context for these significant findings.
8. Further discussion on how these findings might influence current clinical practices and outlining practical subsequent steps is recommended.

Conclusion Section:
9. The conclusion is well-articulated; nonetheless, it should more explicitly highlight how these findings could potentially influence future clinical practices.

Minor:

1. Lines 74-76. This paragraph erroneously belongs to the journal’s template and should not be included in the study.
2. Line 180. Properly cite the subsection: Extracranial Metastasis at the Time of Brain Metastasis Diagnosis.
3. Line 216. Properly cite the subsection: Singular Brain Metastasis.
4. Line 223. Properly cite the subsection: Surgery of Brain Metastasis.
5. Line 229. Properly cite the subsection: Radiotherapy of Brain Metastasis.
6. Line 245. Properly cite the subsection: Lines of Therapy Regimens Before the Diagnosis of Brain Metastasis.
7. Line 274. Properly cite the subsection: Multivariable Survival Analysis.
8. Line 341. Insert the missing period after the reference citation [27, 29].
9. Line 355. Insert the missing period after the reference citation [33].

Author Response

Introduction Section:
1. Lines 50-55, Page 2. It is suggested that the authors expand the background discussion on drug permeability across the blood-brain barrier. For instance, incorporating insights from Corti et al., 2022, may provide a more comprehensive understanding of intracranial efficacy relevant to the treatment of brain metastases.

Thank you for pointing this out. We agree with you on the comment and added a passage to drug permeability across the blood-brain barrier including the nice and relevant review from Conti et al. (2022). Drug permeability at the BBB and its intracerebral efficacy, but also possible changes in this due to the effects of the drugs at the BBB is another important aspect of our research group, where we hope to be able to publish further relevant results in the near future.

Materials and Methods Section:
2. Despite well-documented methods, the inclusion of a flowchart could improve comprehension of the participant selection process.
Please see our response to point 3.

Results Section:
3. Although extensive, the results might be improved by adding additional tables or graphs, especially when presenting complex demographic data. It is essential to clearly justify any missing data in particular categories.

We agree with the reviewer that further explanations of missing data is sensible and have added those in the legend of Table 1.
We also highly agree that tables, graphs, or flowcharts improve readers' understanding and thank the reviewer for this remark. However, we have tried to highlight the most critical findings and information in our manuscript with figures. Together with the new Figure 3 added in the current revision, we present eight different figures and fear that adding even more figures/flowcharts might overwhelm the reader and distract from the main points of our manuscript. Therefore, we are hesitant to go over the current number of figures/flowcharts.
That being said, if the reviewer disagrees with our assessment, we would kindly ask the reviewer for a suggestion on which additional data might benefit most from a representation as a figure/flowchart despite the new Figure 3 and will gladly provide those.

4.Within the Kaplan-Meier plots, it is advisable to clarify the legends, colours, and accompanying statistics to aid result interpretation.

We thank the reviewer for their valuable recommendation and have clarified our figure legends.

5. Lines 187-194, Page 6. A graphical representation could potentially improve the presentation and interpretation of these results.

We thank the reviewer for suggesting ways to improve the presentation of our results and have gladly added a new bar graph to display our findings (Figure 3).

6.Page 7, Lines 202-203. Interpreting these results is challenging. Contrary to the textual description, the accompanying figure and caption do not indicate that the values correspond to the brain metastases group.

Thank you for pointing this out. We agree with you on the comment and therefore added to the attached figure and caption that the values refer to the group of brain metastases.

Discussion Section:
7. The study mentions related works but lacks depth. An expanded account of the referenced studies could enrich the discussion, providing better context for these significant findings.-

Thank you for pointing this out. We agree with you on the comment and therefore deepened the discussion also around the point of BBB permeability and intracerebral efficacy of systemic therapeutics.

8.Further discussion on how these findings might influence current clinical practices and outlining practical subsequent steps is recommended.

Thank you for pointing this out. We agree with you on the comment and therefore added possible practical subsequent steps for the future.

Conclusion Section:
9. The conclusion is well-articulated; nonetheless, it should more explicitly highlight how these findings could potentially influence future clinical practices.

Thank you for pointing this out. We agree with you on the comment and therefore added possible practical subsequent steps for the future.

Minor:
1. Lines 74-76. This paragraph erroneously belongs to the journal’s template and should not be included in the study.
2. Line 180. Properly cite the subsection: Extracranial Metastasis at the Time of Brain Metastasis Diagnosis.
3. Line 216. Properly cite the subsection: Singular Brain Metastasis.
4. Line 223. Properly cite the subsection: Surgery of Brain Metastasis.
5. Line 229. Properly cite the subsection: Radiotherapy of Brain Metastasis.
6. Line 245. Properly cite the subsection: Lines of Therapy Regimens Before the Diagnosis of Brain Metastasis.
7. Line 274. Properly cite the subsection: Multivariable Survival Analysis.
8. Line 341. Insert the missing period after the reference citation [27, 29].
9. Line 355. Insert the missing period after the reference citation [33].

Thank you for pointing this out. We agree with you on the comments and therefore deleted the paragraph incorrectly inserted, properly cited the subsections and inserted the missing period.

Reviewer 3 Report

Comments and Suggestions for Authors

    This manuscript by Curtaz is a retrospective study of over 300  patients with brain metastases..   The study is welll designed and yields important information of prognosis for breast  cancer peitents with brain metastases.  This information  willl be useful for design of future studies in this patient population.

Author Response

Thank you for your comment and review of our study.

Round 2

Reviewer 2 Report

Comments and Suggestions for Authors

Your responses to the previous comments and concerns were thorough and effectively addressed the points raised.